# Evaluating Microbiological Safety, Sensory Quality, and Packaging for Online Market Success of Roasted Pickled Fish Powder

**DOI:** 10.3390/foods13060861

**Published:** 2024-03-12

**Authors:** Aunchalee Aussanasuwannakul, Pisut Butsuwan

**Affiliations:** 1Department of Food Chemistry and Physics, Institute of Food Research and Product Development, Kasetsart University, Bangkok 10903, Thailand; 2Department of Food Processing and Preservation, Institute of Food Research and Product Development, Kasetsart University, Bangkok 10903, Thailand; ifrpisut@ku.ac.th

**Keywords:** roasted pickled fish powder, sensory evaluation, just-about-right, check-all-that-apply, microbiological safety, traditional food products, consumer preferences, packaging optimization, digital marketplace, shelf-life extension

## Abstract

This study explores the pivotal roles of microbiological safety, sensory qualities, packaging efficiency, and consumer preferences in determining the success of roasted pickled fish powder (RPFP) variants in the online marketplace. The comparison of the nutritional composition of the developed RPFP variants with a commercial benchmark reveals significant differences: the protein content in the herbal flavor variant is found to be 28.97%, which is lower than the 40.17% found in the commercial benchmark, while the fat content in the spicy flavor variant is measured at 19.51%, exceeding the 10.60% present in the benchmark. Additionally, the herbal flavor boasts a superior dietary fiber content of 14.23%, highlighting the intricate relationship between nutritional content and sensory attributes, which is critical in online retail, where physical product evaluation is not possible. Our comprehensive approach, evaluating both nutritional and sensory dimensions, introduces a novel perspective to the adaptation of traditional food products for e-commerce, addressing a gap in the literature. Despite this study’s limitations, including a focused market analysis and constrained sample size, our findings offer valuable insights into enhancing product quality and integrity in the digital marketplace, positioning RPFP for competitive success while suggesting directions for future research.

## 1. Introduction

The integration of traditional food products into the digital marketplace represents a significant evolution within the food processing and preservation domain, marking a dynamic confluence of heritage culinary practices and contemporary consumer expectations. Our study, focusing on the development of value-added roasted pickled fish powder (RPFP), strives to harmonize traditional preservation methods with the demands of today’s online consumer preferences. The digital transformation of community-based food enterprises highlights the urgent need to adapt product offerings to cater to a wider and more diverse consumer base, thus enhancing their economic viability and contributing to sustainable local economies [1,2,3,4,5].

Despite significant advances in food technology and e-commerce, there exists a notable gap in our understanding of how innovative food products, especially those emanating from community settings, align with the intricacies of an online-centric market. To bridge this gap, an integrated approach that encompasses a digital presence, insights into consumer behavior, and technological adaptation is essential [6,7]. This study is poised to elevate RPFP to meet high standards of chemical and nutritional quality, ensuring its resonance with the expectations of the online marketplace and leveraging the food product development process through stages crucial for digital market alignment [8,9,10].

Moreover, this research seeks to bolster the economic prospects of community enterprises and empower local producers, offering a model for successful online integration. Scientifically, it aims to enrich the dialogue on food science and product development, illuminating how traditional foods can be reimagined for the digital market. This endeavor gains particular relevance against the backdrop of Thailand’s digital transformation, characterized by high internet penetration and burgeoning online marketplaces, providing a fertile ground for projects aiming to empower women through digital and economic means [11,12].

Plaa-som, a traditional Thai fermented fish dish, serves as a pertinent backdrop for this study, showcasing the versatility and cultural significance of traditional food products. Research into plaa-som’s fermentation process, nutritional enhancements, and adaptability to health-conscious trends lays a foundational understanding for RPFP’s potential in online markets [13,14,15,16,17,18,19]. By detailing the methodologies underpinning RPFP’s development, this study aims to offer insights into the multifaceted approach required for the successful online retailing of traditional food products.

## 2. Materials and Methods

### 2.1. Roasted Flavored Pickled Fish (Plaa-som) Powder Preparation

Fresh pearl spot (*Probarbus jullieni*) fish, ranging from 700 to 1200 g, were procured from a local market and immediately transported in an icebox to the food processing facility of the Freshwater Fish Processing Community Enterprise (Aunt Uan Village No. 3) in Sam Wa East Subdistrict, Khlong Sam Wa District, Bangkok, Thailand. Processing commenced within an hour of purchase. The preparation of the pickled fish (plaa-som) followed traditional methods, as outlined by Onsurathum et al. [20], with subsequent flavoring and dehydration steps adapted from Begum et al. [21]. First, the fish (2 kg) was cleaned under running tap water, gutted, and mixed with glutinous rice, minced garlic, and table salt. This mixture was then divided, sealed in plastic bags, and left to ferment at room temperature for 3 days, emphasizing the importance of hygiene to control sensory quality variation due to the natural fermentation process [22]. Post-fermentation, the fish was steamed for 20 min, seasoned with tamarind juice and salt, fried to reduce moisture, and finally roasted to achieve a moisture content below 10%. The ingredients of each flavor variant of the roasted pickled fish powder are detailed in Table 1.

Following preparation, 50 g samples of roasted pickled fish powder, while still hot (75–80 °C), were packaged into either 10 × 15 cm^2^ laminated foil stand-up pouches made from a laminated film consisting of polyethylene terephthalate, aluminum, and polyethylene with a zip lock or polypropylene cups, and these were immediately sealed, labeled, and stored at room temperature (27 °C) for further analysis. The packaging materials were supplied by TK Supplies and Business Solutions Co., Ltd., Bangkok, Thailand.

### 2.2. Storage Stability

Following storage under ambient conditions (25 °C, 60% relative humidity), the samples underwent qualitative shelf-life testing, including microbiological quality assessments in accordance with local regulations [23] and hedonic testing. In this study, we compared two recipes (herbal vs. spicy flavor) across two time points (0 and 2 months), as well as measuring water activity, moisture content, and pH. The examination was solely centered on the pH and water activity data of the herbal-flavored RPFPs packaged in plastic cups and aluminum pouches over a duration of two months. A commercial benchmark was not utilized for this part of the study, as the aim was to investigate the inherent stability of the newly developed RPFP variants under different storage conditions. This approach was selected to isolate and understand the effects of packaging and time on the product’s stability without an external comparison to commercial products.

### 2.3. Proximate Composition and Nutritional Value

The nutritional composition, including moisture, crude protein, crude lipid, and ash contents, was analyzed in triplicate following AOAC standard procedures [24], with the methods of Sullivan and Carpenter [25] employed for a nutritional labeling analysis. 

In our nutritional evaluation, detailed methodologies were employed to ascertain the nutritional composition and potential health benefits of the developed roasted pickled fish powder (RPFP) variants. The assessment included the determination of the moisture content, protein, fat, ash, total carbohydrates, and energy value using standard procedures as outlined by AOAC International. The protein content was measured using the Kjeldahl method, while the fat content was determined through Soxhlet extraction. The ash content was assessed via incineration in a muffle furnace, and carbohydrates were calculated by difference, subtracting the sum of moisture, protein, fat, and ash from 100. The energy value was estimated using the Atwater system, multiplying protein and carbohydrates by 4 kcal/g and fat by 9 kcal/g.

Furthermore, specific nutritional markers, such as dietary fiber, sodium, and calcium, were quantified using established analytical techniques. Dietary fiber was analyzed using the enzymatic–gravimetric method, and sodium and calcium levels were measured via atomic absorption spectrophotometry, ensuring a comprehensive evaluation of the RPFPs’ nutritional profile. These methods adhere to the rigorous standards set by AOAC International, providing a reliable basis for the nutritional characterization of our RPFP variants.

### 2.4. pH and Water Activity Determination

pH measurements were conducted using a digital pH meter after blending the samples with distilled water. The water activity of the samples was analyzed using a water activity meter (Model LabMaster-aw; Novasina, Zurich, Switzerland).

### 2.5. Microbiological Properties

#### 2.5.1. Yeast and Mold Determination

Yeast and mold in the samples were analyzed following the dilution plating technique described in the FDA’s Bacteriological Analytical Manual (BAM) [26]. First, the samples were prepared by adding 0.1% peptone water to create a 10^−1^ dilution and homogenized. For plating, the spread-plate method was employed using dichloran rose Bengal chloramphenicol agar. Each dilution was plated in triplicate. The plates were incubated in the dark at 25 °C. After a 5-day incubation period, plates containing 10–150 colonies were selected for counting. Colony forming units (CFU) per gram or milliliter were calculated based on average colony counts, rounding off to two significant figures. This enumeration process aids in accurately quantifying yeast and mold presence in food samples, which is crucial for ensuring food safety and quality.

#### 2.5.2. *Escherichia coli* Determination

The determination of *E. coli* followed the method described in the FDA’s BAM [27]. Briefly, the method involved preparing a sample, transferring it to Lauryl tryptose broth for presumptive testing, and incubating it at 35 °C for 24–48 h. Positive results proceeded to BGLB broth, with incubation under the same conditions. Confirmed tests used EC broth, incubated at 44.5 °C for 24 h, with further checks at 48 h. *E. coli* was then isolated using eosin methylene blue agar and identified via colony characteristics and biochemical tests. *E. coli* was quantified using the most probable number estimation method. 

#### 2.5.3. *Staphylococcus aureus* Determination

*S. aureus* was determined in accordance with the FDA’s BAM (CFU/g) [28]. After sample preparation, the sample was diluted, plated on Baird–Parker agar, and incubated at 35–37 °C for 45–48 h. Colonies typical of *S. aureus* were counted and confirmed via coagulase and other ancillary tests. The number of *S. aureus* per gram of food was reported based on colony counts. This method is suitable for foods where more than 100 *S. aureus* cells per gram are expected. 

#### 2.5.4. *Clostridium perfringens* Determination

*C. perfringens* was determined according to the BAM of the FDA [29]. First, a sample (25 g) was homogenized in 225 mL of peptone dilution fluid, followed by serial dilutions and plating on tryptose–sulfite–cycloserine agar. The plates were incubated anaerobically at 35 °C for 24 h. Presumptive *C. perfringens* colonies, which appear black with a white precipitate, were confirmed using further biochemical tests. The count of *C. perfringens* was calculated per gram of sample based on the colony count. This procedure allows for the accurate detection and quantification of *C. perfringens* in food samples. 

#### 2.5.5. *Bacillus cereus* Determination

*B. cereus* was determined according to the method outlined in the FDA’s BAM [30]. The sample was homogenized, followed by plating serial dilutions on mannitol egg-yolk polymyxin agar or Bacara agar. These plates were incubated at 30 °C for 24–48 h. Presumptive *B. cereus* colonies, typically characterized by pink colonies with a lecithinase reaction, were counted. Confirmation involved Gram staining and biochemical tests. The number of *B. cereus* per gram of food was calculated based on colony counts. 

#### 2.5.6. *Salmonella* spp. Determination

The method for determining *Salmonella* spp. was in accordance with ISO 6570-1:2017/Amd.1:2020 [31]. It involved the pre-enrichment of a specific amount of sample in a non-selective broth, selective enrichment, and isolation on differential media. This method ensures the precise detection of *Salmonella* through a series of steps that include incubation at specific temperatures and durations (24 ± 2 h at 35 °C). For the confirmation of presumptive *Salmonella* colonies, biochemical and serological tests were conducted, with a count of *Salmonella* per gram or milliliter (25 g) of sample obtained. 

### 2.6. Sensory Evaluation and Consumer Study

All subjects provided informed consent for inclusion before participating in the study. The study was conducted in accordance with the international guidelines for human research protection, and the methodology was approved by the Research Ethics Committee of Kasetsart University (COE No. COE67/013). Sensory profiling and a consumer analysis of the three RPFP samples were carried out by 60 untrained volunteer panelists (aged 18 years or over) who were habitual consumers of flavored fish products. The panelists were employees of the Institute of Food Research and Product Development, Kasetsart University, Bangkok, Thailand, and they had previously been recruited for and familiarized with the Sensory and Consumer Research Unit’s sensory evaluation protocol. All the panelists completed a questionnaire. The aim of the questionnaire was to ensure a comprehensive understanding of consumer responses. Participant selection was demographically diverse and followed the methodology of Monteiro et al. [32]. The first part of the questionnaire included six short questions on gender, age, education, income, and frequency of exposure to the product. Age category was defined as Generations Z (younger than 25 years), Y (26–43 years), and X (44–58 years), which are related to online shopping behavior [33,34]. The second part involved a sensory evaluation and consumer study of the developed roasted pickled fish samples, with the herbal and spicy flavors evaluated and compared against a commercial benchmark (Kamnan Chul Farm, Rai Nai Chul Kunwong Co., Ltd., Phetchabun, Thailand). The sensory evaluation and consumer study detailed in Section 2.6.1, Section 2.6.2, Section 2.6.3 and Section 2.6.4 encompassed comprehensive analyses. These analyses aimed to assess the sensory attributes of appearance, taste, color, texture, odor, and overall acceptability. Furthermore, consumer preferences were evaluated to understand the acceptance of and purchase intent for the developed samples, in comparison with a commercial benchmark. The study employed structured sensory evaluation methods and consumer surveys to gather insights into the sensory appeal and market potential of the product variants, providing a basis for refining product development and marketing strategies for the online market. The sample was removed from a freshly opened package. Each sample (5 g of RPFP) was coded with three random digits. The sample was presented to the panelists in a clear, press-seal plastic cup to prevent it from absorbing moisture. The sensory evaluation was conducted as a full crossover, where each panelist evaluated each sample in a sequential monadic presentation. Furthermore, the presentation order of the sample product across panelists was balanced, and the assignment of the presentation order to the panelists was randomized.

#### 2.6.1. Acceptance Test

For acceptance testing and acceptability scaling, the consumer appeal of the products was assessed using a rating scale for the degree of liking or disliking. The acceptance scales included the traditional 9-point hedonic scale and just about right (JAR) scale [35]. 

##### Liking or Hedonic Test

The 9-point hedonic scale (1: dislike extremely to 9: like extremely) was used for determining the likability or acceptance of appearance, flavor, and texture and overall acceptance. 

##### Just About Right (JAR) Scale

The 7-point JAR scale was employed to determine the optimal intensity levels of key sensory attributes, including color, sour, salty, spicy, flavor, and texture. These attributes were rated from 1 = “too little” to 5 = “JAR” and 7 = “too much”.

#### 2.6.2. Check All That Apply (CATA) Analysis

On the CATA questionnaire, developed in accordance with the literature [36], the participants selected sensory attributes that they considered relevant from a predetermined list of descriptors. The descriptors used were created from a focus group discussion (FGD), which was conducted in line with that outlined by Nurazizah et al. [37]. The focus group included participants with relevant food product experience collaboratively generating sensory descriptors to understand consumer perceptions of RPFP. Table 2 shows the terms identified by the FGD, together with sensory attributes (no. 1–7) and product positioning (no. 8–14). The CATA question was as follows: “Please check all the attributes that describe the RPFP you have just tried”.

#### 2.6.3. Consumer Purchase Intent

Structured questionnaires were administered to assess the likelihood of consumers purchasing the RPFPs, combining methodologies from Patel et al. [38] and Murillo et al. [39], with consumer purchasing intent evaluated under various hypothetical purchase scenarios. After completing the CATA questionnaire, the subjects were asked the following question: “If the product had a package size of 50 g in a sealed aluminum bag and a price of 59 baht (USD 1.66) on an online store like Shopee, would you buy it?” There were five possible answers: “definitely would not buy”, “probably would not buy”, “may or may not buy”, “probably would buy”, and “definitely would buy”.

#### 2.6.4. Consumer Preference

In the consumer preference test, the participants were asked to compare two of the developed products against each other with the aim of product improvement and parity testing. After they had tried two samples, the first question that they were asked was as follows: “Which sample did you prefer overall?” The possible responses were “preferred the first sample”, “preferred the second sample”, and “no preference/both the same”. They were then asked an open-ended question: “Why did you prefer that sample?” The serving sequences were randomized across the participants, with an equal number of participants receiving either the first sample or the second sample first. 

### 2.7. Statistical Analysis

To compare the means of nutritional composition and consumer liking data among RPFPs, an analysis of variance (ANOVA) was performed, followed by post hoc Tukey’s tests, using XLSTAT 2023.3.0 (1415) software for Macintosh 14.2 [40]. The results are presented as mean ± standard deviation, with a significance threshold set at *p* ≤ 0.05. All analyses were replicated at least three times to ensure reliability. XLSTAT’s CATA data analysis tool was used to automate the analysis of the CATA data, including Cochran’s Q test, a correspondence analysis (CA), a principal coordinate analysis (PCoA), and a consumer clustering analysis (hierarchical cluster analysis). XLSTAT’s CATATIS applied to the JAR data resulted in multivariate and penalty analyses. The CATATIS method for the multivariate analysis of the JAR data resulted in an attribute/product biplot and a penalty analysis. 

Descriptive statistics were used to calculate the mean scores and standard deviations for the sensory attributes assessed via the hedonic scale at 0 and 2 months of storage. Paired *t*-tests contrasted these scores to test the hypothesis of no significant sensory change due to storage. Significant differences were quantified using Cohen’s *d* to illuminate the magnitude of changes. This analysis was replicated thrice for reliability. Data visualization employed box plots to compare sensory attributes and illustrate consumer perceptions’ variability and central tendency over time. Histograms and Kernel Density Estimate (KDE) curves visualized the overall satisfaction scores, indicating the range of the consumer experience with the products.

## 3. Results

### 3.1. Proximate Composition and Nutritional Value

We present a detailed analysis of the proximate compositions and nutritional values of the herbal- and spicy-flavored RPFPs in comparison to those of a commercial benchmark. This analysis, aimed at developing a value-added product for online sales, emphasizes moisture reduction and optimized packaging to extend shelf life. 

Table 3 provides a comparative analysis of the proximate and nutritional compositions of the RPFPs in the two developed flavors (herbal and spicy) and those of the commercial benchmark. The analysis included moisture, protein, fat, ash, total carbohydrate, total energy, and other nutritional parameters. The results revealed distinct differences in the nutritional composition between the developed products and the commercial benchmark. Notably, both developed products exhibited unique profiles in terms of their protein, fat, dietary fiber, sodium, and calcium contents, which are critical for their nutritional and health benefits. The comparative analysis revealed that the protein content of the herbal flavor (28.97%) was higher than that of the spicy flavor (14.48%) but that the protein contents of both were lower than the protein content of the commercial benchmark (40.17%). Thus, although the developed products are rich in proteins, the commercial benchmark is superior in terms of protein content. The fat content of the spicy flavor (19.51%) was higher than that of both the herbal flavor (16.28%) and the commercial benchmark (10.60%), indicating a richer, more energy-dense product. The comparative analysis revealed that the herbal flavor was notably superior in dietary fiber (14.23%) and calcium (422.81 mg/100 g) content compared to the spicy flavor, which exhibited lower dietary fiber (8.22%) and significantly lower calcium (103.20 mg/100 g). This highlights the herbal flavor as potentially offering more health benefits, particularly for bone health and digestive wellness. The sodium content of the herbal flavor (1366 mg/100 g) was higher than that of the spicy flavor (954.88 mg/100 g), which may be a consideration for sodium-restricted diets.

### 3.2. Storage Stability

In the context of enhancing the storage stability of RPFP, with a focus on extending the shelf life for online marketplace viability, this study incorporated a comprehensive analysis of packaging improvements, microbiological safety, and sensory attributes. Such an analysis is vital for ensuring product quality and consumer satisfaction, aligning with the methods described for processing and packaging RPFP.

Selecting the right packaging for the developed RPFPs is crucial for ensuring their quality and safety when sold online. Table 4 provides a comparison of the previously used polypropylene cups with the newly adopted laminated aluminum foil stand-up pouches. Although polypropylene cups might seem a budget-friendly option, their poor barrier properties and vulnerability to shipping damage make them unsuitable for online sales of RPFP. Laminated aluminum foil stand-up pouches, despite their slightly higher upfront cost, offer the following significant advantages:(1)Preserve fresh flavors and aromas, with customers receiving the product at its peak, maximizing satisfaction;(2)Guarantee safe consumption, with a robust barrier minimizing spoilage risk, ensuring product safety;(3)Protect against bumps during transit, arriving in pristine condition;(4)Enhance brand image, with premium packaging reflecting the quality and care put into the product.

**Table 4 foods-13-00861-t004:** Comparison of packaging options for RPFP for online sales: polypropylene cup versus laminated aluminum pouch.

Feature	Polypropylene Cup (PP *)	Laminated Aluminum Pouch (PET/AL/PE **)
Image	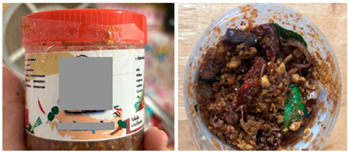	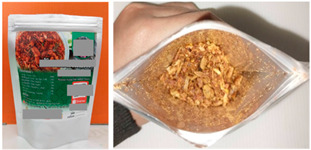
Weight	4.57 g	4.34 g
Dimensions	Diameter = 7 cm; length = 5.5 cm	Width = 10 cm; length = 15 cm
Barrier Properties	Poor; susceptible to moisture, oxygen, and light ingress	Excellent; blocks moisture, oxygen, light, and odors
Shelf Life	Short; prone to flavor degradation and spoilage	Extended; maintains freshness and quality for months
Physical Protection	Vulnerable to dents, punctures, and breakage	Puncture-resistant and sturdy; withstands shipping rigors
Tamper Evident	No; broken seals easily occur	Secure; resealable zip-lock closure prevents contamination
Online Suitability	Not ideal; leakage and breakage risk during shipping	Ideal; protects product during transit and storage
Sustainability	Typically not recyclable or biodegradable	Some pouches are recyclable or contain biodegradable materials
Cost	Lower initial cost	Higher initial cost, but longer shelf life may offset expense

* Polypropylene commonly used in food packaging for its flexibility and resistance; ** A multi-layer film comprising polyethylene terephthalate (PET) for structural integrity, aluminum (AL) as a barrier layer, and polyethylene (PE) for sealing and flexibility.

The RPFP in a laminated bag was further used in analyses of storage stability indices comparing the two recipes. 

Table 5 presents the microbiological safety and stability of the herbal-flavored RPFPs over time. The table details the microbial analyses of the RPFPs packaged in plastic cups and aluminum pouches, assessing the products at the initial (0 months) and final (2 months) time points of the study. The results confirm the microbiological stability of the herbal flavor in both types of packaging. The pH and water activity values in Table 6, which are critical for assessing physical and chemical stability, remained stable across both product flavors and time points, indicating no significant deterioration in product quality. The results confirmed compliance with Thai Department of Medical Sciences (DMSC) standards, showcasing the products’ safety across storage times and packaging types.

The box plots in Figure 1 (top) illustrate the sensory attribute scores of the herbal-flavored RPFPs at 0 and 2 months of storage, showing medians, quartiles, and outliers. Statistical tests revealed no significant differences in appearance, color, odor, taste, texture, or overall liking between storage times, indicating that the sensory attributes remain unaffected for up to 2 months. The histogram and KDE curve in Figure 1 (bottom) highlight the consistency of consumer perceptions of overall quality, with an analysis confirming panel agreement on sensory evaluations across storage times, demonstrating reliable consumer panel performance.

Both the herbal and spicy flavors exhibited robust storage stability, with no discernible difference in their microbiological safety or sensory attributes, suggesting that the product formulations are well suited to long-term storage. The microbiological and sensory evaluation indicated that both types of packaging (plastic cup vs. aluminum pouch) effectively maintained product quality. However, the transition to aluminum pouches is justified by their superior barrier properties. Furthermore, the aluminum pouch’s added benefits of consumer convenience and enhanced physical protection during shipping offer a clear advantage for online sales. In relation to the storage time (0 vs. 2 months), the absence of significant changes in microbiological safety, pH, water activity, and sensory properties over 2 months underscored the excellent storage stability of the RPFP, affirming its suitability as a ready-to-eat product with a prolonged shelf life.

This comprehensive evaluation of storage stability, incorporating microbiological safety, pH, water activity, and sensory attributes, underscores RPFP’s potential as a stable and high-quality product suitable for the online marketplace. The findings provide a solid foundation for the further development and optimization of packaging and storage strategies to enhance product appeal and consumer satisfaction.

### 3.3. Sensory Evaluation and Consumer Study

#### 3.3.1. Demographic Characteristics of Participants

Figure 2 depicts the demographic composition of the participants who took part in the sensory evaluation of the RPFPs. According to the study inclusion criteria, all participants had to be at least 18 years of age. The participants were categorized into different groups to ensure a comprehensive analysis of consumer preferences. The group categorization included gender, age (generation), educational attainment, income level, experience of online shopping, and frequency of online food purchases. Contrary to an equal distribution across these demographics, our findings revealed a diverse composition:Gender: The participant pool comprised a significantly higher proportion of females than males: 78% versus 22%.Generation: The breakdown by generation showed a predominance of Generation Y (53%), followed by Generation X (32%), with Generation Z (15%) being the least represented.Education: Educational levels varied, with 50% holding a bachelor’s degree, 25% having an undergraduate level of education, 17% possessing doctoral qualifications, and 8% having a master’s degree.Income (revenue): The majority, 68%, reported earning less than THB 30,000 (approximately USD 845, based on an exchange rate of THB 35.518 per USD on 7 February 2024). Those with incomes ranging from THB 30,000 to 50,000 accounted for 23%, which translates to approximately USD 845 to USD 1407. A small fraction, 5%, earned above THB 50,000 (approximately USD 1407 or more).Online shopping experience: The majority (87%) of the participants had experience shopping online, with just 13% not having such experience.Frequency of online food shopping: In total, 49% of the participants seldom shopped for food online, whereas 20% did so once a week, 18% once a month, and 13% never did so.

**Figure 2 foods-13-00861-f002:**
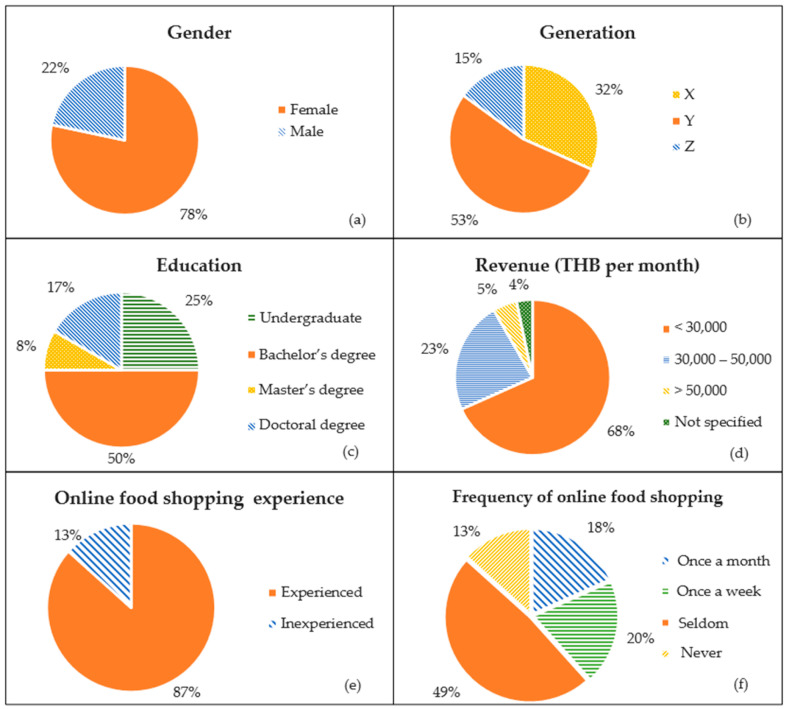
Distribution of the demographic characteristics of the participants (*n* = 60) in the sensory evaluation of RPFP. The pie charts illustrate the percentage distribution across various categories, including (**a**) gender, (**b**) generational groups (X, Y, Z), (**c**) education levels, (**d**) monthly income (in THB), (**e**) online food shopping experience, and (**f**) frequency of online food shopping.

This detailed demographic breakdown not only underlines the diversity of the participant group but also emphasizes the study’s focus on gathering insights from a broad spectrum of consumers who engage with the product category in varying capacities.

#### 3.3.2. Liking

In the sensory evaluation segment focusing on consumer liking for RPFP comprising two product prototypes (herbal flavor: sample code 243; spicy flavor: sample code 697) and a commercial benchmark (sample code: 152), an ANOVA was employed to analyze the preference patterns among these offerings. The results of the ANOVA (*F* = 7.527, Pr > *F* = 0.001) highlighted statistically significant differences in the liking scores among the tested products. Thus, the findings indicate that preferences for at least one of the prototypes markedly differed from the preferences for the commercial standard, underscoring distinct consumer perceptions tied to each product’s sensory profile.

Further examination of these preferences was facilitated by mean charts, which detailed the least square means of the liking scores, effectively revealing the varying degrees of consumer appeal that each product commanded. The commercial benchmark emerged as the most favored product, indicating its broader consumer appeal over the newly introduced herbal and spicy variations.

The differences in preferences are depicted in a bar graph, serving as a representation of the data. As indicated in the bar graph, the commercial benchmark (sample 152) achieved a score of 0.300 ± 0.095, surpassing the scores for the herbal (sample 243) and spicy (sample 697) prototypes, both of which registered a score of −0.150 ± 0.095. This graphical representation not only facilitates an intuitive understanding of the products’ relative standings but also highlights the nuanced consumer inclinations that differentiate between the familiar and the novel (Figure 3).

#### 3.3.3. JAR Analysis

In the JAR analysis, consumer perceptions regarding various sensory attributes of the RPFP samples, encompassing color, sourness, saltiness, spiciness, flavor, and texture intensity, were evaluated. The consumer responses were grouped into three distinct classifications: too little, JAR, and too much. Figure 4 shows the results of the JAR analysis and penalty analysis of the sensory attributes of the RPFPs. 

The evaluation of the color attribute revealed that 87% of the participants deemed the herbal-flavored product (sample code: 243) to possess optimal intensity, indicating a favorable perception that contrasted with that of the commercial benchmark (sample code: 152), which some participants found overly vibrant, and that of the spicy variant (sample code: 697), criticized for its inadequate color intensity. In the assessment of sourness, the benchmark was considered JAR by 67% of the participants, a higher proportion than the herbal (243) and spicy (697) samples, which received JAR ratings from 57% and 52% of the participants, respectively. These findings point to variation in consumer acceptance of sourness levels, with a notable segment finding the sourness intensity of the benchmark insufficient, while the herbal and spicy variants were perceived by some as excessively sour.

The distribution of the JAR ratings for saltiness demonstrated relative uniformity across all samples, with perceived adequacy ranging from 68 to 77%. However, the attribute of spiciness was particularly divisive for the spicy sample (697), with 88% of consumers finding its spiciness overly intense, in stark contrast to the more moderate perceptions of the herbal (sample code: 243) and benchmark (152) samples.

In terms of flavor, both the benchmark (152) and herbal (243) samples were found to have JAR intensity by 72% and 70% of consumers, respectively. The spicy sample (697), however, was considered to have an appropriate flavor intensity by only 42% of consumers, with a significant portion finding it too mild. Texture was identified as a challenge primarily for the spicy sample, with 80% of consumers indicating an excessively intense texture, unlike the benchmark sample, which was found JAR by 82% of the participants.

In Figure 4 (right side), the penalty analysis is depicted, showing the drop in overall liking on a 9-point hedonic scale against the percentage of consumers who rated sensory attributes as “too low” or “too high”. This analysis determined the impact of deviations from the ideal intensity of sensory attributes on overall liking. It revealed that attributes perceived as too low in intensity incurred greater penalties to overall product liking. For example, an insufficient flavor intensity resulted in a significant deduction of 1.5 points on the 9-point hedonic scale, as reported by approximately 45% of consumers. This was particularly critical for the spicy sample (697), where overwhelming spiciness was predicted to reduce the hedonic score by 0.3 points, corroborating the findings from the JAR analysis. Conversely, although the JAR analysis indicated texture and sourness as problematic for the herbal (243) and spicy (697) samples, only the herbal variant was noted to receive a significant penalty for these attributes being overly intense. In contrast, the benchmark (152) sample exhibited significant reductions in overall liking due to insufficient texture and sourness, while saltiness intensity was a key factor for both prototypes, with nearly 20% of consumers indicating its inadequacy, leading to an approximate deduction of 0.9 points on the hedonic scale.

#### 3.3.4. CATA Analysis

The CATA methodology was utilized to evaluate the sensory attributes of the RPFP samples comprising the two prototypes designated as “herbal flavor” (sample code: 243) and “spicy flavor” (sample code: 697), as well as a “commercial benchmark” (sample code: 152). The participants selected attributes from a predefined list that accurately reflected the sensory characteristics of each sample. The attributes considered were as follows: “tangy”, “salty”, “spicy”, “light and airy”, “crumbly roasted granules”, “aromatic with herbal notes”, “intensely flavorful”, “100% roasted pickled fish”, “non-greasy”, “bone free”, “no MSG, no preservatives”, “protein packed”, “shelf stability”, and “packaging integrity”.

##### Cochran’s Q Test

Cochran’s Q test revealed statistically significant differences in the perception of these attributes across the samples, highlighting their distinct sensory profiles (Table 7). For example, the “spicy” flavor was predominantly associated with “tangy” and “spicy” attributes, distinguishing it from the other samples. Meanwhile, the “herbal” flavor was noted for its “aromatic with herbal notes” and “light and airy” texture, and the “commercial benchmark” was often linked to “crumbly roasted granules” and “intensely flavorful” qualities, indicating the unique sensory profile of the benchmark compared to that of the prototypes.

##### CA

A subsequent CA provided a visual depiction of how each sample correlated with specific sensory attributes, illustrating the unique positioning of each product variant within the sensory space (Figure 5). The results of the CA underscored the alignment of the commercial benchmark with “crumbly roasted granules” and “intensely flavorful” attributes, setting it apart from the prototypes, which were more closely related to the attributes “spicy”, “tangy, “and “aromatic with herbal notes”.

##### PCoA

The PCoA further dissected the relationships between the sensory attributes and consumer perceptions, demonstrating that attributes such as “protein packed”, “shelf stability”, and “packaging integrity”, along with “100% roasted pickled fish” and “non-greasy”, played a significant role in shaping overall liking (Figure 6). This analysis captured a substantial portion of the variability in sensory perceptions across the samples.

##### Consumer Clustering Analysis

A consumer clustering analysis of the sensory perception of the RPFP samples was systematically conducted using a hierarchical cluster analysis and the CLUSCATA method. This methodological approach allowed for the segmentation of consumers into distinct groups, each reflecting a unique set of sensory experiences with the product samples. The analysis, visualized through a dendrogram in Figure 7, delineated two primary consumer classes, each characterized by distinctive perceptual profiles toward the attributes of the fish powder samples. 

The exploration of these classes through biplots provided further insights into the perceptual differences between them. Class 1, marked by a homogeneity score of 0.572, displayed a relatively uniform perception of the products, contrasting with Class 2, which had a homogeneity score of 0.495. The biplots revealed that Class 1 was closely aligned with the sensory attributes identified in the CA, with samples 243 and 697 being distinctly separated from sample 152 on the first axis, indicating shared attributes, such as “tangy”, “spicy”, “100% roasted pickled fish”, “shelf stability”, “no MSG, no preservatives”, and “bone free”. Conversely, Class 2 demonstrated a clear differentiation in perception, particularly distinguishing sample 697 from samples 243 and 152, highlighting unique attribute associations for each group.

The biplots for Class 1 and Class 2 detail the variance explained on the first and second axes and illustrate the sensory perception differences among the samples within each class (Figure 8). Class 1′s biplot emphasizes the shared sensory characteristics between the prototypes, and Class 2′s biplot underscores the distinct sensory profiles that set the spicy flavor apart.

Complementing the clustering analysis, demographic insights were obtained to provide a comprehensive understanding of each cluster’s composition (Table 8). Cluster 1 predominantly consisted of females (77%), with Generation Y (50%) accounting for a significant proportion, followed by Generations X (23%) and Z (27%). This cluster also showed higher educational attainment, with the majority holding bachelor’s degrees (69%) and most earning below THB 30,000 (approximately USD 845, based on an exchange rate of THB 35.518 per USD on 7 February 2024) per month (73%). Their online food shopping experiences varied, with a notable portion shopping seldom (46%) or weekly (19%). 

In contrast, Cluster 2 contained a slightly higher proportion of females (79%) and leaned toward older generations, with Generations Y (56%) and X (38%) predominating. This cluster also had a diverse educational background, with a significant number of participants below the undergraduate level (32%) and above master’s degree (21%). The income distribution closely mirrored that in Cluster 1, although a slightly higher percentage reported earnings above THB 50,000 (approximately USD 1407, based on an exchange rate of THB 35.518 per USD on 7 February 2024). The online shopping experience was notably high, with the majority shopping online seldom (50%) or monthly (24%).

The detailed demographic analysis underscores the diversity within the consumer clusters, highlighting the importance of considering various consumer attributes in product development and marketing strategies to cater to the different sensory preferences revealed through the CATA analysis.

#### 3.3.5. Consumer Purchase Intent

The assessment of consumer purchase intent in relation to RPFP offered valuable insights into market receptivity. The potential market demand for RPFP was determined by evaluating consumer intentions to purchase under a defined scenario. The participants were presented with a hypothetical offering: a 50 g package of the product, sealed within an aluminum pouch, priced at THB 59 (approximately USD 1.66, based on an exchange rate of THB 35.518 per USD on 7 February 2024), and available for purchase on an online platform, such as Shopee (shopee.co.th). This scenario was designed to mirror a realistic online shopping experience, aiming to gauge consumer purchase decisions in a digital marketplace context.

The collected data indicated a diverse range of purchase intentions across the different samples. For the herbal-flavored sample (243), a notable number of participants (67%) expressed a tentative “maybe buy” decision, suggesting a moderate level of interest that might be converted into actual purchases with appropriate marketing strategies or product adjustments. Conversely, the commercial benchmark sample (152) elicited a less enthusiastic response, with only 43% of respondents indicating a “maybe buy” intention, highlighting potential areas for improvement or differentiation to enhance its market appeal. The spicy-flavored sample (697) encountered some resistance, with 49% of the survey participants leaning toward a “probably won’t buy” decision, indicating significant hesitance that could impact its market penetration and necessitate the reconsideration of its flavor profile or marketing approach.

These findings are systematically represented in a bar graph (Figure 9), which illustrates the distribution of purchase intentions across the evaluated samples. In this visual representation, consumer responses are categorized into definitive levels of purchase interest, ranging from “definitely would purchase” to “definitely would not purchase”, thereby providing a clear overview of consumer predisposition toward each product variant.

#### 3.3.6. Consumer Preference 

The consumer preference study aimed to elucidate the preference patterns for the RPFP samples, specifically samples 243 (herbal flavor) and 152 (benchmark), across various age demographics. This segment of the study was structured to identify any significant trends in consumer preferences that could inform product positioning and marketing strategies. 

The participants were segmented into three age groups, namely, younger than 25 years (Generation Z), 26–43 years (Generation Y), and 44–58 years (Generation X), to assess how preferences might vary across different generations. The findings revealed that sample 152 (benchmark) was consistently favored across all age groups due to its mellow flavor profile, accessible texture, and subdued spicy and citrus notes, which collectively appealed to a broad spectrum of participants. For sample 243 (herbal flavor), while its robust flavor profile and authentic orange fish aroma, underscored by visible fish pieces, were appreciated for adding credibility to its origin, certain aspects, such as an overly intense flavor profile, pronounced sour notes, and a persistent fish aroma, were perceived negatively by some participants. Conversely, sample 152 was commended for its balanced flavor and fine, crumbly texture, although it was critiqued for potentially overemphasizing sweetness and saltiness and lacking the distinctive orange fish aroma that might have differentiated it further from competing products.

These insights are visualized in a bar graph (Figure 10), which provides a detailed comparison of consumer preferences for the RPFP samples across the designated age groups. This graphical representation facilitates a nuanced understanding of how each sample was received by different generational cohorts, highlighting the specific attributes that contributed to the overall preference for sample 152 over sample 243.

## 4. Discussion

Our study underscores the pivotal role of microbiological safety, sensory qualities, packaging efficiency, and alignment with consumer preferences in the online marketplace. In particular, the significant variations in nutritional composition between our developed RPFP variants and the commercial benchmark highlight the necessity of balancing traditional food product qualities with consumer health considerations. The protein content of the herbal flavor (28.97%) being lower than that of the commercial benchmark (40.17%) echoes findings from similar studies, indicating a potential gap in nutritional enhancement that could bridge traditional methods and commercial standards [41,42]. Additionally, the trade-off between flavor enhancement and nutritional cost, exemplified by the higher fat content in the spicy flavor (19.51%), demands careful consideration in product development for health-conscious consumers [43,44].

A noteworthy observation in our study is the discrepancy between the percentage of pickled fish added to each formula, as reported in Table 1, and the protein content of the finished products, as indicated in Table 3. This discrepancy may be attributed to variations in the protein content of the pickled fish, influenced by factors such as the fish type, size, age, and pickling process. Additionally, processing losses during production, including heat treatment and dehydration, could account for differences in the expected and observed protein content. These insights underline the complexity of achieving consistent nutritional profiles in food products and underscore the importance of considering these factors in product development and analysis.

Central to our findings is the discovery that the herbal flavor’s superior dietary fiber content (14.23%) and the spicy flavor’s heightened spiciness intensity underscore the intricate relationship between nutritional content and sensory attributes. This relationship is critical in online retail, where physical product evaluation is not possible, and purchasing decisions heavily rely on product descriptions and visual representations. Maintaining consistent sensory qualities can foster consumer trust and satisfaction, which are crucial for online sales success [45,46]. The emphasis on dietary fiber and spiciness opens avenues for targeted marketing strategies, addressing specific consumer health interests and taste preferences [47,48].

Our comprehensive evaluation of RPFP variants demonstrated their adherence to stringent microbiological safety standards, confirming their suitability for online retail. The meticulous analysis of microbial stability, including the assessment of pH levels and water activity over the storage period, underscores the effectiveness of our preservation techniques. These findings are crucial, as microbiological safety not only affects consumer health but also influences product reputation and trustworthiness in the online marketplace. Ensuring the microbiological integrity of RPFP variants through optimized processing and preservation methods reflects our commitment to delivering safe, high-quality food products to consumers.

The selection of packaging materials played a pivotal role in maintaining the sensory and nutritional integrity of the RPFP variants throughout the supply chain to the consumer. Our study highlights the transition from traditional polypropylene cups to laminated aluminum foil stand-up pouches, driven by the need for enhanced barrier properties, physical protection, and consumer convenience. These packaging solutions significantly contributed to extending the shelf life of the product by offering superior protection against oxygen, moisture, and microbial contamination, enhancing the overall product appeal and reinforcing brand image in the competitive online marketplace.

Our approach, evaluating RPFP variants through both nutritional and sensory lenses for online retail success, introduces a novel perspective in the field. This comprehensive assessment, including microbial safety and storage stability, offers a robust framework to ensure product quality in the digital marketplace. By addressing the literature gap in traditional food product enhancement for e-commerce, our study provides valuable insights into adapting traditional products to meet modern consumer demands [49,50].

However, our study’s limitations, including the constrained sample size and focused market analysis, necessitate caution in generalizing these findings. Future research could broaden the scope of traditional products and market environments explored, validating our insights across a wider context. Additionally, delving into the long-term effects of storage on sensory and nutritional quality will enrich our understanding of product shelf-life and consumer satisfaction over time, further contributing to the field’s knowledge base [51].

By navigating the complexities of online food retail with a multifaceted approach that prioritizes product integrity, sensory and nutritional expectations, and consumer preferences, RPFP stands well positioned for success in the competitive digital marketplace. This strategy not only addresses current market needs but also anticipates future consumer trends, establishing RPFP as a forward-thinking, consumer-centric offering.

## 5. Conclusions

Our study illuminated the critical importance of microbiological safety, sensory qualities, packaging efficiency, and consumer preferences in the online marketplace, particularly highlighting the intricate balance required between traditional food product attributes and modern consumer health expectations. The comprehensive evaluation conducted on roasted pickled fish powder (RPFP) variants demonstrated that variations in nutritional composition, specifically in protein and dietary fiber content, between the herbal and spicy flavors in comparison with a commercial benchmark significantly influence consumer trust and satisfaction in digital retail environments. This analysis underscores the potential for targeted marketing strategies that leverage these attributes to cater to specific consumer health interests and taste preferences.

Moreover, our investigation into the storage stability of RPFP variants revealed their robustness in maintaining microbiological safety and sensory attributes over time, underscoring the efficacy of optimized packaging solutions, particularly the transition to laminated aluminum foil stand-up pouches, in preserving product quality and extending shelf life. This focus on packaging efficiency not only enhances consumer appeal but also aligns with sustainability considerations crucial for online sales success.

Our novel approach in examining both the nutritional and sensory dimensions of RPFP variants for online retail success offers valuable insights for enhancing traditional food products in the e-commerce landscape. This dual focus contributes to closing the gap in the current literature, providing a robust framework for ensuring product quality and integrity in the digital marketplace.

However, it is crucial to acknowledge the limitations of our study, including the constrained sample size and the focused nature of our market analysis. These constraints may limit the generalizability of our findings, suggesting a need for caution in extrapolating our results to broader contexts. Future research should aim to expand the scope of traditional products and market environments explored, further validating our insights. Additionally, investigating the long-term effects of storage on sensory and nutritional quality will deepen our understanding of the factors that influence product shelf-life and consumer satisfaction, offering valuable directions for ongoing research in this domain.

In conclusion, by navigating the complexities of online food retail with a comprehensive approach that integrates product integrity, sensory and nutritional expectations, and consumer preferences, RPFP is strategically positioned for success in the competitive digital marketplace. Nevertheless, recognizing and addressing the limitations of this study are essential for advancing our knowledge and effectively meeting the dynamic demands of the digital age.

## Figures and Tables

**Figure 1 foods-13-00861-f001:**
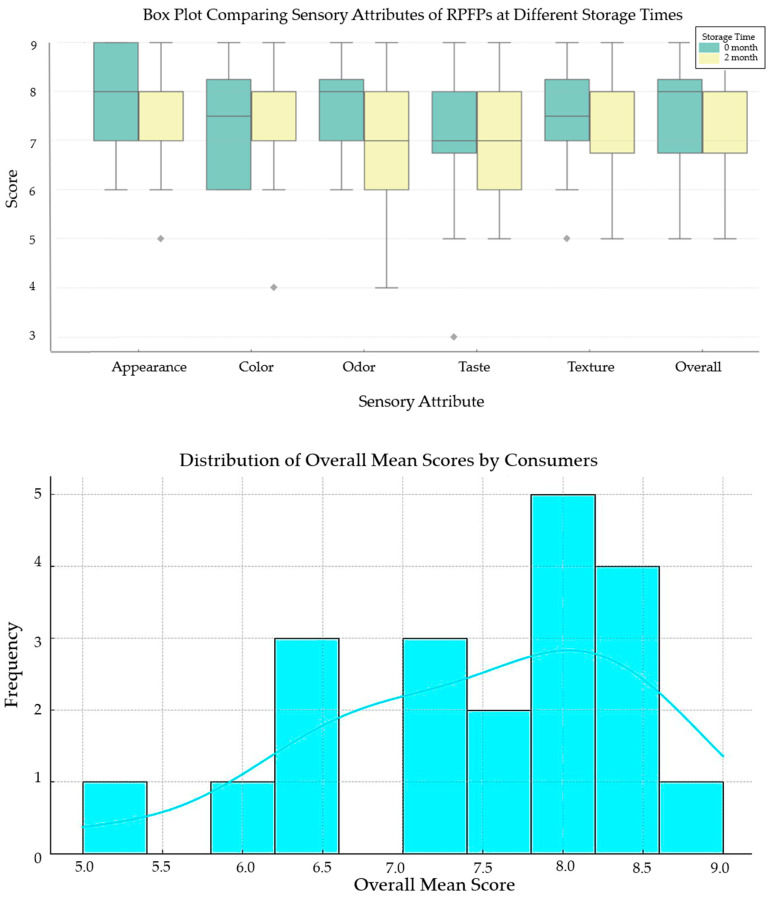
(**Top**) Box plots comparing sensory attributes (appearance, color, odor, taste, texture, and overall liking) of herbal-flavored RPFPs at 0 and 2 months of storage, displaying consumer ratings’ variability and central tendency on a 9-point hedonic scale. (**Bottom**) Distribution of overall mean scores by consumers, with a histogram and KDE curve showcasing the central tendency and spread of overall satisfaction scores, highlighting the range of consumer experiences with the products over time.

**Figure 3 foods-13-00861-f003:**
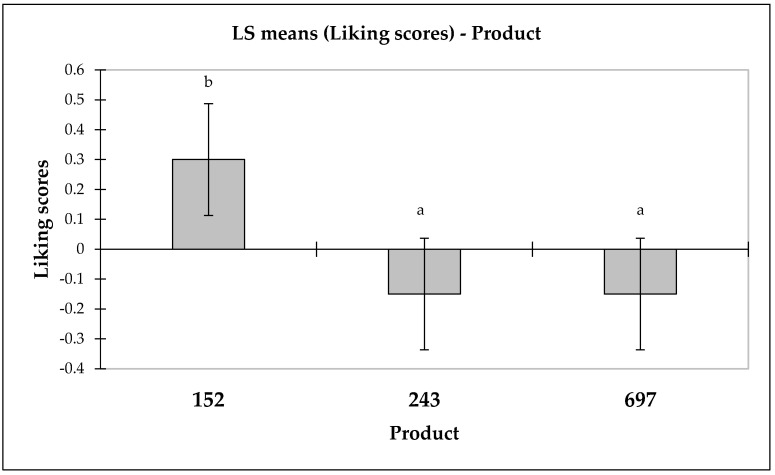
Comparative analysis of consumer liking scores for the RPFP variants, detailing the least square means, along with standard errors for the liking scores of sample 152 (commercial benchmark), sample 243 (herbal flavor), and sample 697 (spicy flavor). Distinctions in statistical significance are denoted by the different letters positioned above the respective bars.

**Figure 4 foods-13-00861-f004:**
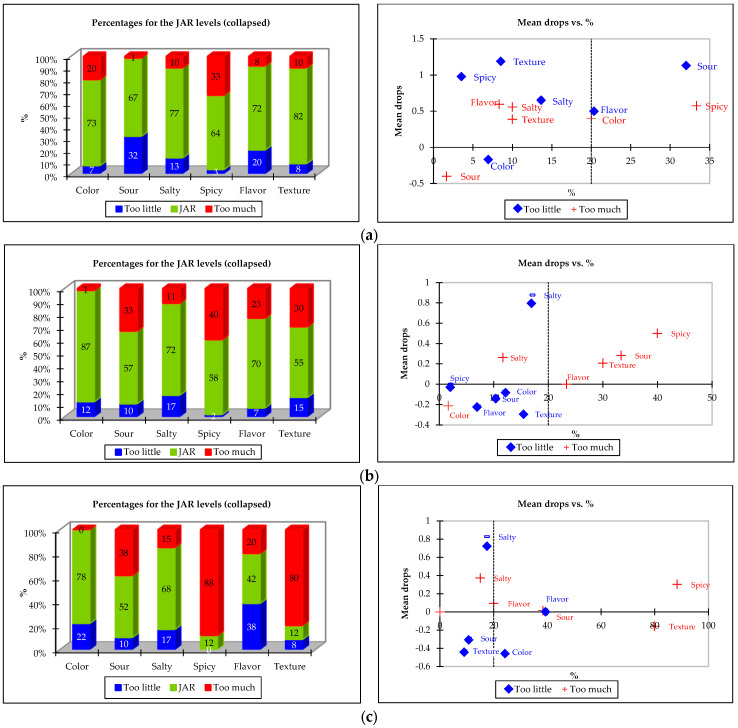
Distribution of the JAR ratings for the sensory attributes of the RPFP samples (**left side**), depicting consumer responses for color, sourness, saltiness, spiciness, flavor, and texture intensity, categorized as too little, JAR, and too much for each product sample, and the penalty analysis on the 9-point hedonic scale for each sensory attribute being too low or too high in intensity (**right side**): (**a**) commercial benchmark (152), (**b**) herbal flavor (243), and (**c**) spicy flavor (697).

**Figure 5 foods-13-00861-f005:**
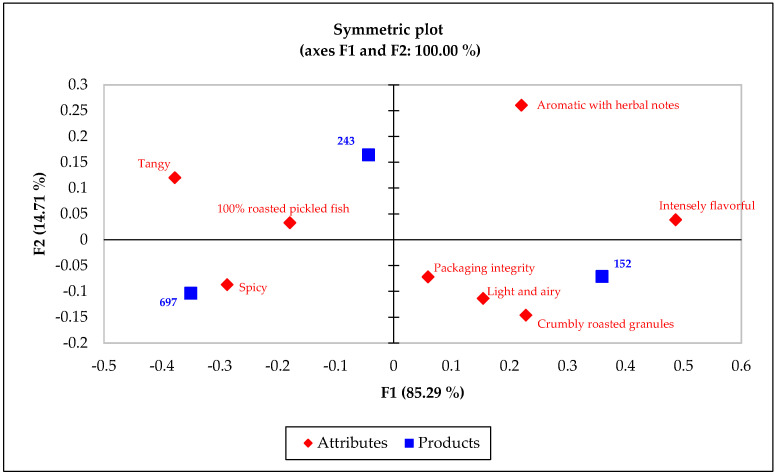
Biplot from CA, exhibiting relationship between CATA attributes and RPFP samples.

**Figure 6 foods-13-00861-f006:**
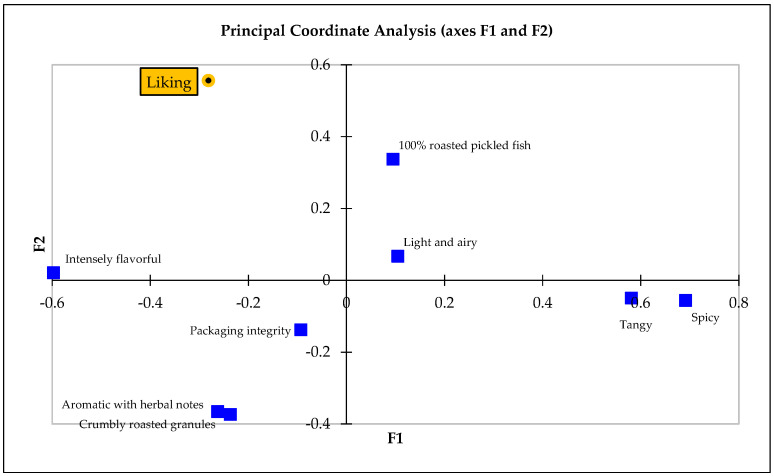
PCoA two-dimensional map depicting interplay between sensory attributes and consumer liking.

**Figure 7 foods-13-00861-f007:**
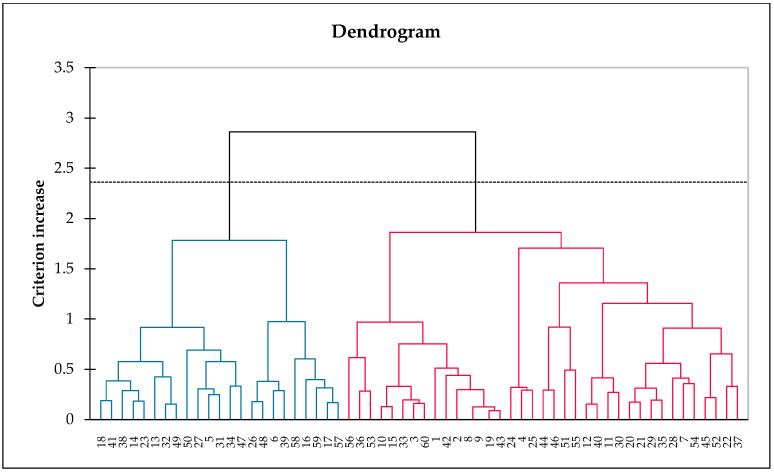
Dendrogram from CLUSCATA analysis, showing segmentation of consumer groups into two main classes based on RPFP sensory perceptions. Clusters are differentiated by color: blue indicates preliminary groupings, red denotes distinct classes as marked by the dotted line.

**Figure 8 foods-13-00861-f008:**
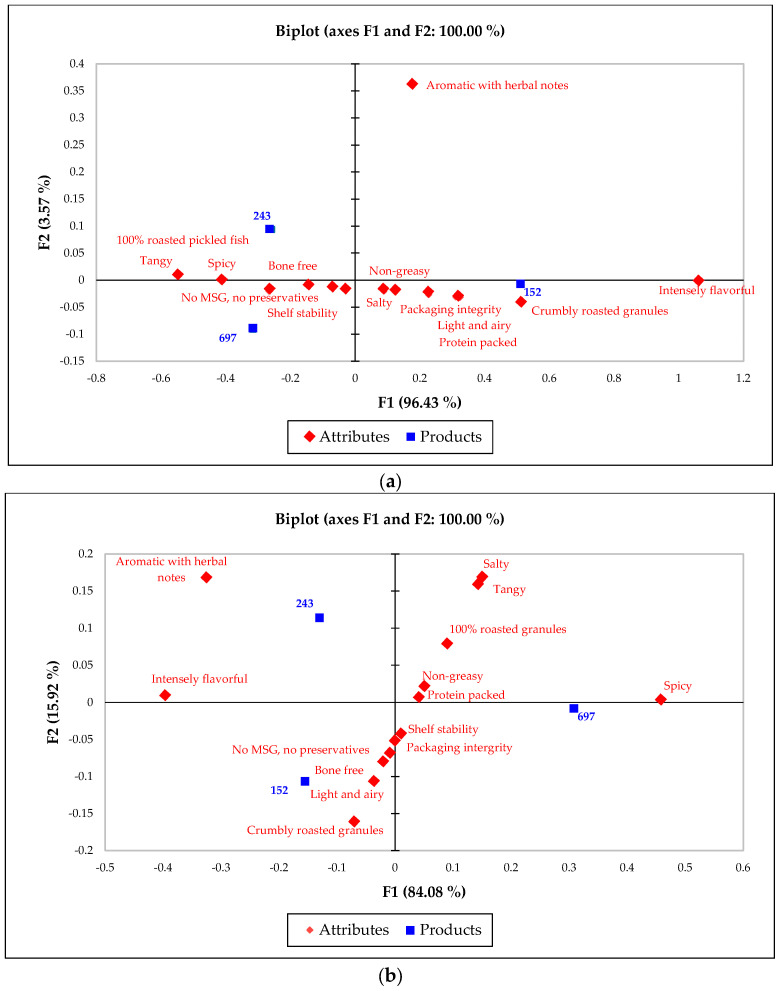
A biplot of the classes in the CLUSCATA analysis, displaying variance explained on the first and second axes: (**a**) sensory perception differences in Class 1 of the RPFP sample; (**b**) the differences in attributes among the RPFP samples in Class 2.

**Figure 9 foods-13-00861-f009:**
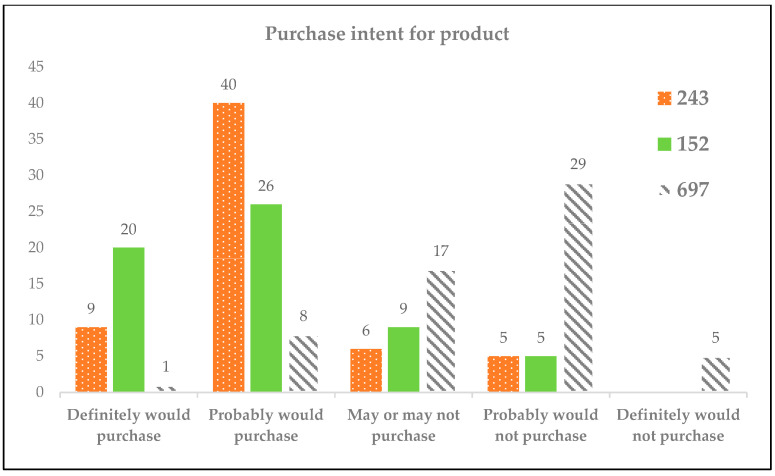
Distribution of purchase intentions for RPFP (samples 243, 152, and 697) among consumers, with these classified as “definitely would purchase”, “probably would purchase”, “may or may not purchase”, “probably would not purchase”, and “definitely would not purchase”.

**Figure 10 foods-13-00861-f010:**
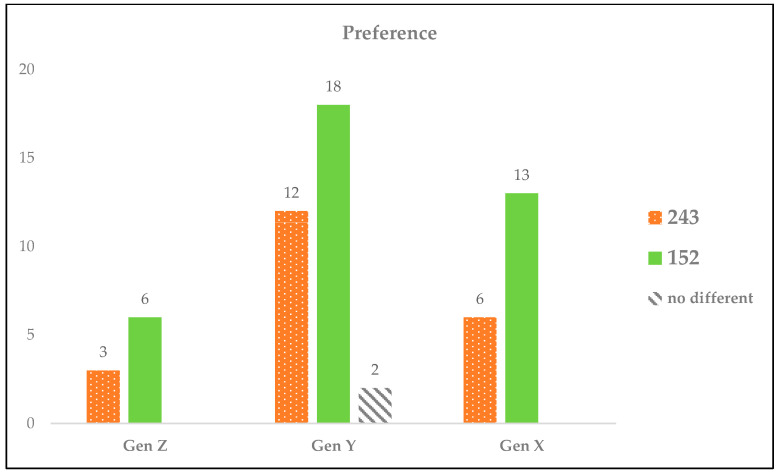
Comparative generational preferences for RPFP, illustrating the preferences among Generations Z (<25 years), Y (26–43 years), and X (44–58 years) for samples 152 (benchmark) and 243 (herbal flavor), including indications of no significant preference.

**Table 1 foods-13-00861-t001:** Ingredients of roasted pickled fish powder variants.

Code	Product	Brand	Sample	Ingredients
152	Commercial benchmark	Kamnan Chul farm	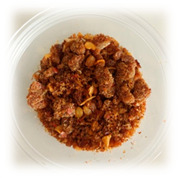	Pickled fish (36.90%), fried garlic (16%), fried shallot (16%), granulated sugar (9%), tamarind juice (8%), chili powder (2.8%), kaffir lime leaves (1.6%), seasoning powder (0.7%)
243	Roasted pickled fish (herbal flavor)	Developed product	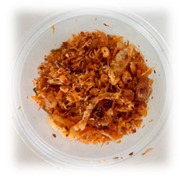	Pickled fish (60%), dried bird’s eye chili (15%), fried shallot (10%), fried herd mix (garlic, galangal, lemon grass) (5%), tamarind juice (5%), salt (4%), seasoning powder (1%)
697	Roasted pickled fish (spicy flavor)	Developed product	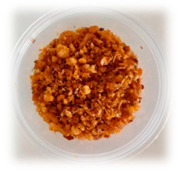	Pickled fish (60%), dried bird’s eye chili (15%), fried shallot (10%), fried garlic (5%), tamarind juice (5%), salt (4%), seasoning powder (1%)

**Table 2 foods-13-00861-t002:** Sensory attributes and non-sensory (product positioning) attributes based on the results of the FGD.

No.	Attribute	Description
1	Tangy	A zesty, acidic quality typical in fermented foods
2	Salty	The flavor attribute from sodium ions, essential in flavoring and preservation
3	Spicy	A heat sensation from capsaicinoids or similar compounds
4	Light and airy	Textural characteristic of a low-density, porous food matrix
5	Crumbly roasted granules	A brittle, uneven texture resulting from roasting and the Maillard reaction
6	Aromatic with herbal notes	A complex scent profile from volatile compounds in herbs
7	Intensely flavorful	A strong, multidimensional taste and aroma profile
8	100% roasted pickled fish	Product of lacto-fermented fish and thermal processing
9	Non-greasy	Absence of excess surface lipids, enhancing mouthfeel
10	Bone free	A uniform texture due to the removal of calcified elements
11	No MSG, no preservatives	The quality of having no added MSG or synthetic preservatives
12	Protein packed	A high protein level, significant for nutrition labeling
13	Shelf stability	Extended preservation of product quality and freshness
14	Packaging integrity	Use of spoilage-preventing technologies in packaging

No.

**Table 3 foods-13-00861-t003:** Comparative nutritional compositions of the RPFPs (herbal and spicy variants vs. a commercial benchmark).

Analysis Item	Herbal Flavor	Spicy Flavor	Commercial Benchmark
Moisture (%)	15.49 ± 0.12 ^c^	13.72 ± 0.09 ^b^	11.56 ± 0.11 ^a^
Protein (%) (factor 6.25)	28.97 ± 0.13 ^b^	14.48 ± 0.38 ^a^	40.17 ± 0.72 ^c^
Fat (%)	16.28 ± 0.16 ^b^	19.51 ± 0.88 ^c^	10.60 ± 0.08 ^a^
Ash (%)	6.74 ± 0.13 ^b^	4.03 ± 0.17 ^a^	8.60 ± 0.15 ^c^
Total carbohydrate (%)	32.52 ± 0.06 ^b^	48.26 ± 0.40 ^c^	29.07 ± 0.16 ^a^
Total energy (kcal/100 g)	392.48 ± 3.06 ^b^	426.55 ± 1.98 ^c^	372.36 ± 3.87 ^a^
Energy from fat (Kcal/100 g)	146.52 ± 1.37 ^b^	175.59 ± 2.96 ^c^	95.40 ± 1.88 ^a^
Saturated fat (%)	6.16 ± 0.03 ^a^	7.72 ± 0.10 ^b^	-
Cholesterol (mg/100 g)	87.87 ± 0.75 ^b^	42.07 ± 0.50 ^a^	-
Dietary fiber (%)	14.23 ± 0.04 ^b^	8.22 ± 0.07 ^a^	-
Sugars (%)	5.63 ± 0.10 ^a^	7.14 ± 0.15 ^b^	-
Sodium (mg/100 g)	1366 ± 4.73 ^b^	954.88 ± 1.54 ^a^	-
Vitamin A (beta carotene) (µg/100 g)	646.56 ± 4.68 ^b^	565.78 ± 6.85 ^a^	-
Vitamin B1 (mg/100 g)	0.12 ± 0.01 ^a^	0.11 ± 0.00 ^a^	-
Vitamin B2 (mg/100)	0.12 ± 0.01 ^a^	0.10 ± 0.00 ^a^	-
Calcium (mg/100 g)	422.81 ± 4.68 ^b^	103.20 ± 2.32 ^a^	-
Iron (mg/100 g)	4.18 ± 0.03 ^a^	4.62 ± 0.09 ^a^	-

^a–c^ Means that entries within the same column that have the same superscript or no superscript are not significantly different (*p* > 0.05; *n* = 3).

**Table 5 foods-13-00861-t005:** Comprehensive microbiological analysis of herbal-flavored RPFPs across storage conditions and times.

	Criteria	Plastic Cup 0 Months	Plastic Cup 2 Months	AL Pouch 0 Months	AL Pouch 2 Months
Yeasts and molds (CFU/g)	<100	<10	<10	<10	<10
*Escherichia coli* (MPN/g)	<3	<3	<3	<3	<3
*Staphylococcus aureus* (CFU/g)	<10	<10	<10	<10	<10
*Clostridium perfringens* (CFU/g)	<100	<10	<10	<10	<10
*Bacillus cereus* (CFU/g)	<1000	<10	<10	70	<10
*Salmonella* spp. (/25 g)	N.D. *	N.D.	N.D.	N.D.	N.D.

* N.D. = Not detected.

**Table 6 foods-13-00861-t006:** Stability of pH and water activity of RPFP: a comparative analysis of flavors over time.

	Herbal Flavor		Spicy Flavor	
Parameter	0 Months	2 Months	0 Months	2 Months
pH	4.67	4.65	4.33	4.32
water activity	0.67	0.67	0.598	0.60

**Table 7 foods-13-00861-t007:** Cochran’s Q test results for the sensory attributes of the RPFP samples.

Attributes	*p*-Values	152	243	697
Tangy	<0.0001	0.200 ^a^	0.533 ^b^	0.533 ^b^
Salty	0.895	0.317 ^a^	0.300 ^a^	0.300 ^a^
Spicy	<0.0001	0.450 ^a^	0.600 ^a^	0.833 ^b^
Light and airy	0.029	0.383 ^a^	0.250 ^a^	0.250 ^a^
Crumbly roasted granules	<0.0001	0.567 ^b^	0.317 ^a^	0.317 ^a^
Aromatic with herbal notes	0.00	0.367 ^b^	0.433 ^b^	0.150 ^a^
Intensely flavorful	<0.0001	0.700 ^c^	0.433 ^b^	0.150 ^a^
100% roasted pickled fish	0.002	0.400 ^a^	0.567 ^b^	0.567 ^b^
Non-greasy	0.549	0.567 ^a^	0.517 ^a^	0.517 ^a^
Bone free	0.135	0.650 ^a^	0.583 ^a^	0.583 ^a^
No MSG, no preservatives	0.513	0.250 ^a^	0.217 ^a^	0.217 ^a^
Protein packed	0.819	0.183 ^a^	0.167 ^a^	0.167 ^a^
Shelf stability	0.368	0.450 ^a^	0.417 ^a^	0.417 ^a^
Packaging integrity	0.069	0.333 ^a^	0.267 ^a^	0.267 ^a^

The table presents the *p*-values for each attribute, with multiple pairwise comparisons using the critical difference (Sheskin) procedure. ^a–c^ Significant differences are indicated by different letters for each attribute across the product samples (sample codes: 243, 697, and 152 for the herbal flavor, spicy flavor, and benchmark, respectively).

**Table 8 foods-13-00861-t008:** Demographic distribution of consumer clusters in sensory evaluation of RPFP.

	%	
	Cluster 1	Cluster 2
**Gender**		
Female	77	79
Male	23	21
**Generation**		
X	23	38
Y	50	56
Z	27	6
**Education**		
Undergraduate	15	32
Bachelor’s degree	69	35
Master’s degree	4	12
Doctoral degree	12	21
**Revenue (THB per month)**		
<30,000	73	65
30,000–50,000	27	21
>50,000	0	9
Not specified	0	6
**Frequency of consumption (per week)**		
0 day	8	9
1 day	50	44
2 days	31	18
3 days	4	15
4 days	4	9
5 days	0	0
6 days	0	0
7 days	3	5
**Online food shopping experience**		
Experienced	77	94
Inexperienced	23	6
**Frequency of online food shopping**		
Once a month	12	24
Once a week	19	21
Seldom	46	50
Never	23	6
**Preference**		
243	27	41
152	69	56
No difference	4	3
**243 overall liking score**		
1–5	12	6
6–9	88	94
**152 overall liking score**		
1–5	4	9
6–9	96	91
**697 overall liking score**		
1–5	12	6
6–9	88	94

## Data Availability

The original contributions presented in the study are included in the article, further inquiries can be directed to the corresponding author.

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
