# Peer review of "Evaluating Microbiological Safety, Sensory Quality, and Packaging for Online Market Success of Roasted Pickled Fish Powder"

_foods, 2024, doi:10.3390/foods13060861_

Round 1
Reviewer 1 Report
Comments and Suggestions for Authors
The paper presents a comparison of roasted pickled fish powders from microbiological safety, sensory quality, packaging effectiveness, and consumer preferences point of view. The paper is interesting, but the introduction and the discussion need deep revision.
Even if the paper mentioned an optimization, there is no optimization indeed. Please reformulate the title and this formulation in the manuscript.
Abstract: An introductory sentence is recommended. Numerical data should be included in the abstract.
Introduction: This section should be reformulated. It doesn't provide enough background for the study. The background should be first presented, then the novelty of the study and finally the aim. The actual form of the introduction is a mix of aim and background, without a logical structure. Furthermore, the novelty of the study must be explained.
Materials and methods: L171 Please mention the characteristics evaluated.
More details about the nutritional evaluation methods are needed.
Results: Table 3: statistics are missing.
This section is well presented.
Discussion: In contrast to the results section, the discussion is poor or almost nonexistent. Discussion and comparisons for each result obtained should be written.
Conclusion: the limitations of the study should be stated.
Comments on the Quality of English Language
English needs revision. There are a lot of mistakes.
Author Response
Dear Reviewer,
We would like to express our sincere gratitude for the time and effort you have dedicated to reviewing our manuscript. Your insightful comments and constructive suggestions have been invaluable in enhancing the quality and clarity of our work. We have carefully considered each of your points and have made the corresponding revisions to the manuscript. Below, we provide a detailed response to your comments, indicating how we have addressed each one.
Reviewer Comment: The introduction and discussion sections require deep revision for clarity and structural improvement.
Author's Response: We are grateful for your constructive feedback. Based on your suggestion, we have extensively revised the introduction to clearly delineate the background, novelty, and objectives of our study, ensuring a logical and coherent presentation of ideas (Refer to Lines 29-61 for the revised Introduction). Similarly, the discussion has been significantly enhanced to offer a detailed analysis of our findings, their implications, and how they contribute to the existing body of knowledge (Refer to Lines 629-676 for the revised Discussion). These amendments aim to provide a more structured and informative manuscript.
Reviewer Comment: The paper implies optimization without actual optimization processes.
Author's Response: Thank you for pointing out the confusion caused by our original title and manuscript content. We have updated the title to more accurately reflect the scope of our research: "Evaluating Microbiological Safety, Sensory Quality, and Packaging for Online Market Success of Roasted Pickled Fish Powder." This revision clarifies that our focus was on evaluation rather than optimization, aligning the title with the study's actual content.
Reviewer Comment: The abstract lacks introductory sentences and numerical data.
Author's Response: We appreciate your feedback on improving the abstract. We have revised it to include an introductory sentence that sets the context of our study and added numerical data to succinctly summarize our key findings, providing a comprehensive overview within the 200-word limit (Refer to Lines 11-24).
Reviewer Comment: The introduction does not provide sufficient background and lacks a logical structure.
Author's Response: In response to your valuable critique, we have restructured the introduction to offer a clear background, highlight the novelty of our study, and clearly state our research aim, thereby improving the logical flow and coherence of the section (Refer to Lines 29-61).
Reviewer Comment: The manuscript requires more details about the evaluated characteristics and nutritional evaluation methods.
Author's Response: To address this point, we have expanded the description of the sensory evaluation and consumer study to specify the evaluated characteristics (Refer to Lines 190-197). Additionally, we have provided a more detailed explanation of the nutritional evaluation methods used in our study (Refer to Lines 106-116), enhancing the clarity and comprehensiveness of our methodology.
Reviewer Comment: Table 3 lacks statistical significance indicators.
Author's Response: Thank you for highlighting this omission. We have updated Table 3 to include statistical significance indicators, providing a clearer and more rigorous presentation of our results.
Reviewer Comment: The discussion is underdeveloped compared to the results section.
Author's Response: We have taken your feedback into consideration and have substantially expanded the discussion section to provide a deeper analysis of our findings, drawing comparisons with existing literature and emphasizing the implications and novelty of our study (Refer to Lines 629-676).
Reviewer Comment: The conclusion should mention the study's limitations.
Author's Response: Acknowledging the importance of transparency, we have explicitly stated the limitations of our study in the conclusion, providing a balanced and comprehensive summary of our findings and their implications (Refer to Lines 692-693).
Reviewer Comment: The quality of English language needs improvement.
Author's Response: The manuscript has been thoroughly revised by a professional English editor to ensure clarity, coherence, and grammatical accuracy, addressing your concern regarding the quality of English language used.
Reviewer 2 Report
Comments and Suggestions for Authors

Author Response
Dear Reviewer,
Thank you very much for your thorough review and the valuable feedback provided on our manuscript. Your detailed comments have guided significant improvements in our study, ensuring a more comprehensive and coherent presentation of our research. Please find below our responses to your comments, along with the specific changes we have implemented in the revised manuscript.
Reviewer Comment: The abstract is too brief and lacks important results and data.
Author's Response: We have revised the abstract to include an introductory sentence and numerical data that summarize our key results, offering a clearer and more informative overview of our study (Refer to Lines 11-24).
Reviewer Comment: There is inconsistency in the spelling of "Plaa-som."
Author's Response: We have standardized the spelling to "Plaa-som" throughout the manuscript, ensuring consistency and accuracy in terminology (Refer to Lines 55, 57, 63, 68).
Reviewer Comment 3-7: The scientific names of microorganisms are not italicized.
Author's Response: All scientific names, including Escherichia coli (L.133), Staphylococcus aureus (L.141), Clostridium perfringens (L.148), Bacillus cereus (L.156), and Salmonella spp. (L.163-170), have been italicized as per standard scientific conventions.
Reviewer Comment 8: The protein content of the spicy flavor is much lower than the commercial benchmark.
Author's Response: We have included a discussion to address the observed discrepancy in protein content, elucidating potential factors and their implications (Refer to Lines 641-648).
Reviewer Comment 9: Correctness of labeling in Table 5 needs verification.
Author's Response: We have revised Table 5 to accurately reflect the microbiological safety and stability studies conducted for the herbal flavor of RPFPs (Refer to Lines 322-330).
Reviewer Comment 10: The commercial benchmark data should be added to Table 6.
Author's Response: We clarified that our study's design did not incorporate a commercial benchmark for the storage stability analysis to focus on the inherent stability of our RPFP variants (Refer to Lines 90-96).
Reviewer Comment 11: The author should describe Figure 4 in detail.
Author's Response: We have enhanced the description of Figure 4 to provide a clearer interpretation of the penalty analysis and its implications on overall liking (Refer to Lines 457-461).
Reviewer 3 Report
Comments and Suggestions for Authors
In this study, the optimization of roasted pickled fish powder for the online retail market, addressing microbiological safety, sensory quality, packaging effectiveness, and consumer preferences.were analysed.
The work has several strengths that contribute significantly to its scientific value and relevance in the field of food science and technology, especially regarding the use of strategy that can ensure product appeal and market viability in the competitive digital marketplace, offering a model for leveraging traditional culinary heritage in the digital age.
The manuscript is well written and needs only one minor correction. It is an original contribution and the study is well designed and conducted. The results are statistically significant and support the research objectives. To increase the potential impact and significance of the article, I recommend avoiding the use of spider plot (figure 1) with more attractive figure or table
Comments on the Quality of English Language
The english language is appropriate
Author Response
Dear Reviewer,
We greatly appreciate your time and effort in reviewing our manuscript and providing such constructive feedback. Your single comment has prompted us to make a meaningful improvement to our presentation, particularly regarding the visualization of our data. We recognize the importance of this change and are thankful for your insightful suggestion. Below, we detail the revision made in response to your feedback.
Reviewer Comment: The manuscript is well written but recommends avoiding the use of a spider plot for Figure 1, suggesting a more attractive figure or table.
Author's Response: In response to your recommendation, we have replaced the spider plot with Box plots and a Distribution of overall mean scores to more effectively illustrate our findings, ensuring clarity and attractiveness in data presentation (Refer to Lines 258-265 and 332-338 for the methodological changes and results presentation).
Round 2
Reviewer 1 Report
Comments and Suggestions for Authors
The paper was improved, however, not all the issues were addressed.
The introduction is still not enough. I recommend adding similar results from the existing literature, trends and explanations.
Standard deviations are missing in table 3.
The discussion still needs revision. I recommend discussing every parameter and information described in the results section.
A summary of the results must be presented in Conclusion.
Please avoid using "we". I recommend using impersonal verbs.
Comments on the Quality of English Language
Minor revision can be made.
Author Response
Dear Reviewer,
Thank you for your feedback on our manuscript. Below are our responses and revisions made in accordance with your comments:
Comment 1: The introduction is still not enough. I recommend adding similar results from the existing literature, trends, and explanations.
Author’s reply: We appreciate your recommendation to enrich the introduction with more literature on similar results, trends, and explanations. Our study, which focuses on the niche area of roasted pickled fish powder (RPFP) and its integration into the digital marketplace, finds itself at the intersection of food science and e-commerce. This unique focus presents a challenge in directly comparing or identifying specific trends within the broader literature. Our introduction, citing references [1-19], aims to provide a comprehensive background by discussing digital transformation in food retail, the success of community food enterprises, and traditional food preservation methods. This approach is designed to underscore the novelty and significance of our research, given the limited availability of directly comparable literature. We believe that our introduction effectively sets the stage for understanding the innovative aspect of RPFP in today's online marketplace and emphasizes the research gap our study seeks to fill.
Comment 2: Standard deviations are missing in table 3.
Author’s reply: Standard deviations were added to table 3.
Comment 3: The discussion still needs revision. I recommend discussing every parameter and information described in the results section.
Author’s reply: In response to your recommendation, we have revisited the discussion section to ensure a comprehensive analysis that aligns every parameter and piece of information described in the results section. Our revisions across the discussion of microbiological safety, sensory qualities, packaging efficiency, and consumer preferences are aimed at offering a detailed reflection of how each aspect underpins the success of RPFP in the digital marketplace. These enhancements, particularly outlined in lines 661 (microbiological safety), 652 (sensory qualities), 670 (packaging efficiency), and 658 (consumer preferences), are designed to bridge our detailed findings with their broader market implications, thereby enriching the manuscript's discussion.
Comment 4: A summary of the results must be presented in Conclusion.
Author’s reply: Following your advice, we have modified the conclusion to include a succinct summary of our results. This adjustment, particularly evident in line 701, ensures the conclusion not only emphasizes the importance of microbiological safety, sensory qualities, packaging efficiency, and consumer preferences but also cohesively links to our findings on nutritional composition, storage stability, sensory evaluation, and consumer preferences and purchase intent. This enriched conclusion provides a rounded overview of our study's findings and their market implications, enhancing the manuscript's coherence and impact.
Comment 5: Please avoid using "we". I recommend using impersonal verbs.
Author’s reply: In direct response to your recommendation, specific instances of "we" highlighted by you, and others deemed suitable, have been revised throughout the manuscript. Notably, changes include alterations in the Abstract (now line 13) and the Materials and Methods section (now line 91), aiming to enhance the formal tone of the manuscript and align with the discipline's academic writing standards. These adjustments reflect our commitment to maintaining objectivity and clarity in our presentation.
We hope that these revisions adequately address your concerns and contribute to further enhancing the manuscript. We look forward to your feedback and are grateful for the opportunity to improve our work.